# Antioxidative and Immunomodulatory Potential of the Endemic French Guiana Wild Cocoa “Guiana”

**DOI:** 10.3390/foods10030522

**Published:** 2021-03-03

**Authors:** Elodie Jean-Marie, Didier Bereau, Patrick Poucheret, Caroline Guzman, Frederic Boudard, Jean-Charles Robinson

**Affiliations:** 1Laboratoire COVAPAM, UMR Qualisud, Université de Guyane, 97300 French Guiana, France; elodie.jean-marie@univ-guyane.fr (E.J.-M.); didier.bereau@univ-guyane.fr (D.B.); 2Qualisud, University Montpellier, Avignon Université, CIRAD, Institut Agro, IRD, Université de La Réunion, 34090 Montpellier, France; patrick.poucheret@umontpellier.fr (P.P.); caroline.guzman@umontpellier.fr (C.G.); frederic.boudard@umontpellier.fr (F.B.)

**Keywords:** cocoa, Amazonia, antioxidant, anti-inflammatory, fermentation

## Abstract

Guiana is a little-known and endemic variety of cocoa (*Theobroma cacao* L.), native to French Guiana. No data were available regarding its chemical composition and biological properties; therefore, a study was necessary, using Forastero as a reference. To exemplify biological activities of the cacao species, cocoa extracts were evaluated by antioxidant (DPPH, FRAP, ORAC) and anti-inflammatory assays. Our results showed that raw Guiana presented equivalent DPPH and FRAP activities, but a 1.3-fold higher antioxidant activity (1097 ± 111.8 μM ET/g DM) than Forastero (838.5 ± 67.8 μM ET/g DM) in ORAC assay. Furthermore, the impact of fermentation (under four conditions: unfermented, two days, four days and six days of fermentation) on Guiana cocoa beans composition and health properties was also studied. Indeed, fermentation, a key step necessary to obtain the taste and color of chocolate, is generally known to alter bean composition and modulate its health benefits. At six days, the fermentation process led to a nearly 25% lower antioxidative capacity in various assays. Moreover, in inflammation-induced macrophage assays, Guiana and Forastero unfermented extracts induced a 112% stimulation in TNF-α production, and a 56.8% inhibition of IL-6 production. Fermentation altered the cocoa composition by diminishing bioactive compounds, which could be responsible for these biological activities. Indeed, after six days of fermentation, compounds decreased from 614.1 ± 39.3 to 332.3 ± 29 mg/100 g DM for epicatechin, from 254.1 ± 14.8 to 129.5 ± 20.7 mg/100 g DM for procyanidin B2 and from 178.4 ± 23.5 to 81.7 ± 2.9 mg/100 g DM for procyanidin C1. The similar composition and the equivalent or higher antioxidant activity of Guiana leads us to propose it as an alternative to Forastero.

## 1. Introduction

Cocoa designation *Theobroma cacao* L., belonging to the Malvaceaes family and Theobroma genus, is a plant native to Neotropical rainforests with more than 10 known existing varieties [1]. Our study mainly focused on two of them: Guiana and Forastero. Originally from the Lower Amazon, Forastero would have been cultivated in Brazil and in Venezuela. Since 1825, this variety has been called “foreigner”, corresponding to a cocoa variety introduced in Trinidad and Tobago [2]. Guiana is native to the Amazon Basin and the Guiana Shield. It has been described for a long time as a subspecies of Lower Amazon Forastero, despite their auto-incompatibility and considerable morphological differences in the fruits. Indeed, Guiana have rougher pods and smaller beans than Forastero. Nevertheless, it is now considered as a full-fledged variety [1,3,4].

The chemical composition of several varieties of cocoa has already been studied. High fat contents account for more than 40–57% of the bean weight, and the majority of lipids, including stearic, oleic, palmitic and linoleic acids, were found in varying amounts depending on variety [5,6]. Sterols, fibers, minerals and methylxanthine compounds such as theobromine, theophylline and caffeine were relevant in the bean compositions [6]. Whatever the variety, cocoa is well-known for its rich polyphenol contents (from 12 to 18% in seed weight) [7]. These compounds are recognized for their positive health effects on the cardiovascular and nervous systems, and through their anticarcinogenic and antioxidant properties [8,9]. Antioxidant (AO) capacity, due to its high phenolic content, was proved in various studies [10] to involve monomeric catechins and more complex by-products, using in vitro [11,12], ex vivo, and in vivo assays [13].

Cocoa is also well-known to stimulate the immune system by presenting anti-inflammatory (AI) activities. Indeed, the immune system is based on two complementary pathways: the first is an innate non-specific response, and the second is an adaptative response involving specific reactions to antigens. The innate response is characterized by an inflammation process. In many pathologies, particularly metabolic, the inflammatory state can become chronic and harmful for the cellular tissues and their functions. Thus, the use of anti-inflammatory agents may be of great interest. The cocoa anti-inflammatory effects have already been underlined due to its polyphenol’s effects on mediators such as tumor necrosis factor alpha (TNF-α) and interleukin 1 (IL-1) and 6 (IL-6) [14]. These cocoa compounds would inhibit the activation and maturation of T and B lymphocytes and would reduce cytokine production [15].

Considering chocolate as the most consumed food in the world, its basic ingredient (cocoa beans) must undergo several stages of processing in order to acquire its typical aroma. Among these, fermentation and roasting play a major role in the formation of flavors. However, these steps also induce chemical modifications, engendering alterations in seed composition and biological properties [16,17,18]. The main objectives of our study were: (i) to assess for the first time Guiana cocoa chemical composition (methylxanthines and polyphenols) and biological properties (specifically antioxidant and immunomodulatory characteristics); and (ii) to determine the impact of Guiana cocoa fermentation on the latter properties.

## 2. Materials and Methods

### 2.1. Plant Material

Guiana and Forastero cocoa pods were harvested in Sinnamary, French Guiana, from the CIRAD cocoa station (GPS data: 5°20′05.8″ N 52°57′14.9″ W). The pods were discarded, and only cocoa beans (with shells) were used for the next steps. They were divided into 2 groups: those to be unfermented and those to be fermented at different degrees of fermentation. In the case of the unfermented group, the pulp of fresh beans was removed by pectinase enzyme (Sigma Aldrich, St. Louis, MO, USA) without physical alteration of the bean. For the other group, beans were fermented for 2, 4 and 6 days in plastic boxes. The choice of these fermentation times was based on the fact that the samples taken on the second day were at the alcoholic fermentation stage, and those taken on the fourth day were at the acetic fermentation stage. The choice to stop the process at 6 days of fermentation is explained by the fact that this corresponds to the maximum duration traditionally chosen to obtain the Forastero commercial cocoa. Then, each group was dried in oven at 60 °C, cooled with nitrogen, and ground into a spray powder.

### 2.2. Extraction Procedure

The Cocoa extract (CE) was realized as follows. Freeze-dried cacao powder was degreased by maceration with n-hexane (Carlo Erba, Milan, Italy) overnight. Following this, this mixture was immerged in an ultrasonic bath for 10 min at 130 kHz. Then, the mixture was centrifuged (5000× *g*, 4 °C, 10 min) and filtrated. This procedure was repeated five times and all supernatants were collected, then evaporated to measure fat content. The fat-reduced cocoa powder was then extracted following the same procedure with a mixture of acetone/water/formic acid (70:30:0.1, *v*/*v*/*v*) (Carlo Erba, Milan, Italy) three times. Acetonic extracts were evaporated, then dissolved in 50% methanol (Carlo Erba, Milan, Italy) at a concentration of 40 g/L. Finally, CEs were stored in the dark at −20 °C until analysis. These extractions were realized in triplicates.

### 2.3. Total Polyphenol Contents

The total polyphenol content (TPC) of the CE was determined spectrophotometrically according to Arnous et al. (2002), with some modifications [19]. Briefly, in a hemolysis tube, 2300 μL of ultrapure water, 100 μL of previously prepared extract at different concentrations, 150 μL of Folin–Ciocalteu’s reagent (Carlo Erba, Milan, Italy), and 450 μL of 20% Na_2_CO_3_ (Carlo Erba, Milan, Italy) were mixed. After 2 h in the dark at room temperature, the absorbance of the blue color was measured at 750 nm against a blank sample with a UV–VIS 50 Bio Varian spectrophotometer. A calibration curve with gallic acid (Panreac, Barcelona, Spain), as standard was produced. The TPC was expressed as the milligram equivalent of gallic acid per gram of dry matter (mg GAE/g DM). All TPC measurements were performed in triplicate.

### 2.4. Total Flavonoid Contents

The total Flavonoid content (TFC) was quantified after derivation with dimethylaminocinnamaldehyde (DMACA) (Fluka, Swiss) in 1N HCl acid methanolic medium (Carlo Erba, Milan, Italy), which conducted to a colored carbonium ion at 640 nm. The assay was realized following the method of Arnous et al. (2002) with some modifications [19]. In a hemolysis tube, 400 μL of CE at different concentrations were added, followed by 2000 μL of DMACA reagent. Incubation was realized in dark conditions at room temperature for 10 min. Then, absorbances were measured at 640 nm with a UV–VIS 50 Bio Varian spectrophotometer. A calibration curve with catechin (Sigma Aldrich, St. Louis, MO, USA) as standard was produced. Results were expressed as the microgram catechin equivalent per gram of dry matter (μg CE/g DM). All TFC measurements were performed in triplicate.

### 2.5. Evaluation of In Vitro Antioxidant Activity by Chemical Assays

#### 2.5.1. 2,2-Diphenyl-Lpicrylhydrazyl (DPPH) Assay

DPPH assay was performed according to the method described by Brand-Williams et al. [20]. This procedure allows the measurement of AO activity by following the reaction between antioxidant and a stable radical, 2,2-diphenyl-lpicrylhydrazyl (DPPH•) (Aldrich chemistry, St. Louis, MO, USA). Indeed, in a hemolysis tube, 3900 μL of a DPPH methanolic solution (0.1 mM) and 100 μL of CE at different concentrations were mixed. After 2 h in the dark at room temperature, the absorbance was measured by a UV–VIS 50 Bio Varian spectrophotometer at 515 nm. Trolox (Acros Organics, Geel, Belgium) was used as antioxidative reference, and results were expressed in μmol of Trolox equivalent per gram of dry matter (μmol TE/g DM). A calibration curve was realized with various concentrations of Trolox (0, 320, 640, 960, 1280 and 1600 μM), and the IC_50_ and TE concentrations were determined with the following equations:(1)$${\mathrm{IC}}_{50}=\frac{50}{\mathrm{linear}\;\mathrm{regression}\;\mathrm{slope}\;\mathrm{of}\;\mathrm{Trolox}\;\mathrm{curve}}$$
(2)$$\mathrm{TE}\;\mathrm{in}\;\mu \mathrm{mol}\frac{\mathrm{TE}}{g}\mathrm{DM}=\left(({\mathrm{IC}}_{\left(50\;\mathrm{trolox}\right)}/{\mathrm{IC}}_{\left(50\;\mathrm{extract}\right)}\times 1000\right)\times \mathrm{extraction}\;\mathrm{yield})/250.29$$

All DPPH measurements were performed in triplicate.

#### 2.5.2. Ferric Reducing Antioxidant Power (FRAP) Assay

This procedure is based on the ferric reducing ability of each standard solution according to Benzie and Strain, with some modifications [21]. Briefly, in hemolysis tube, 100 μL of CE at different concentrations, 300 μL ultrapure water and 3000 μL of the FRAP reagent were mixed at 37 °C in the dark. The FRAP reagent was freshly prepared by mixing the three following solutions in the reported ratio 10:1:1 (*v*/*v*/*v*): 2,4,6-Tri(2-pyridyl)-s-triazine (TPTZ) at 10 mM (Fluka analytical, Swiss), FeCl_3_ (Fischer Scientific, Waltham, MA, USA) at 20 mM in 40 mM hydrochloric acid (HCl) (Carlo Erba, Milan, Italy), and 300 mM sodium acetate buffer (Carlo Erba, Milan, Italy), (pH 3.6). Readings of the absorption maximum at 593 nm were taken after 30 min in dark conditions at room temperature using a UV–VIS 50 Bio Varian spectrophotometer. The Fe^2+^ solution was used to perform the calibration curves. Results were expressed as μmol equivalents of Fe^2+^ per gram of dry matter (μmol Fe^2+^ Equation/g DM). All FRAP measurements were performed in triplicate.

#### 2.5.3. Oxygen Radical Absorbance Capacity (ORAC) Assay

The ORAC assay was carried out according to a procedure described by Morel et al. (2012), with some modifications [22]. For ORAC assessment, all reagents were prepared in 75 mM phosphate buffer (pH 7.4), as well as the original cocoa extracts that were diluted as appropriate. Peroxyl radical was generated using 2,2′-azobis(2-methylpropionamidine) dihydrochloride (AAPH) (Acros Organic, Geel, Belgium). In a 96-well plate, 25 μL of CE and 150 μL fluorescein (117 nM) (Honeywell Fluka, Swiss) were pre-incubated for 10 min at 37 °C. Then, 50 μL of AAPH solution (108.5 mM) was rapidly added using a multichannel pipette. The microplate was immediately placed in a Varian Cary Eclipse reader and the fluorescence was recorded every 3 min for 140 min at excitation and emission wavelengths of 485 and 520 nm, respectively. Five calibration solutions using Trolox (Acros Organics, Geel, Belgium) (0–200 μM, final concentration) as the antioxidant were also carried out in each assay. Regression equations between the area under the fluorescence decay curve (AUC) and antioxidant concentration were calculated for all the Trolox standard solutions. All the reaction mixtures were prepared in triplicate, and at least three independent assays were performed for each sample. Final ORAC values are expressed as Trolox equivalents using the standard Trolox curve calculated for each assay. Results were expressed as μmol of Trolox equivalent/milligram of dry matter (μmol TE/mg DM). All ORAC measurements were performed in triplicate.

### 2.6. Evaluation of Immunomodulatory Activity by Cellular Assays

#### 2.6.1. Macrophage Culture

Murine macrophage cells J774-A1 were grown in RPMI 1640 medium (Sigma Aldrich, St. Louis, MO, USA) containing 10% heat-inactivated fetal bovine serum (Sigma Aldrich, St. Louis, MO, USA), and 1% of penicillin/streptomycin (Sigma Aldrich, St. Louis, MO, USA) in a humidified incubator at 37 °C and 5% of CO_2_.

#### 2.6.2. Cell Viability Assay

The 3-(4,5-dimethylthiazol-2-yl)-5-(3-carboxymethoxyphenyl)-2-(4-sulfophenyl)-2H-tetrazolium (MTS) (Sigma Aldrich, St. Louis, MO, USA) and phenazine methosulfate (PMS) (Sigma Aldrich, St. Louis, MO, USA) reduction assay was based on the following procedure. Murine macrophages were plated at a density of 0.6 million cells/mL on 96-well plates in 100 μL Hanks’ balanced salt solution (HBSS) (Sigma Aldrich, St. Louis, MO, USA), for 16 h at 37 °C, 5% CO_2_. Then, a pre-treatment of 100 μL of CE proceeded for 24 h. The cells were incubated with 50 μL of MTS/PMS for 4 h at 37 °C. The amount of MTS formazan product was determined by measuring absorbance using a micro-plate reader (Molecular Devices) at a test wavelength of 540 nm.

#### 2.6.3. Dosage of Nitric Oxide (NO)

In 24-well plates, 500 μL of murine macrophages at 1 million cells/mL were plated during 2 h at 37 °C, 5% CO_2_. After forming a cellular mat by adhesion, cells were pre-treated with CE (12.5, 25 and 50 μg/mL) for 4–24 h. Cell inflammation was induced with 100 μL lipopolysaccharide (LPS) *Escherichia coli* serotype 55B5 (Sigma Aldrich, St. Louis, MO, USA) at 100 ng/mL and murine interferon gamma (IFNγ) (Thermo Fischer Scientific, Villebon sur Yvette) at 10 UI/mL for 18 h. The supernatant of the pre-treated macrophages (at 4 and 24 h) was harvested. The quantification of NO was obtained by measuring the absorbance of 100 μL of cell culture supernatant revealed with Griess’s reagent (100 μL) at 540 nm with a micro-plate reader (Molecular Devices). The NO inhibition (%) was determined spectrophotometrically with a calibration curve of NaNO_2_ (1.56 to 100 μM). All NO measurements were performed in triplicate.

#### 2.6.4. Cytokine Measurements by ELISA

The supernatant of the pre-treated macrophages (4 and 24 h) was harvested and stored at −80 °C until analysis. Tumor necrosis factor alpha (TNF-α), and interleukin-6 (IL-6) of the 4 and 24 h pre-treated samples were quantified via ELISA assay kit (eBioscience). Samples for TNF-α analysis were measured with the diluted supernatant (1/10) for all the samples, and for IL-6, supernatants were diluted by 1/10 for unstimulated and 1/50 for stimulated samples with a micro-plate reader (Molecular Devices) at 540 nm. With calibration curves (0 to 1000 pg/mL of TNFα or 0 to 500 pg/mL of IL-6), results are expressed as the % stimulation of TNF-α production and % inhibition of IL-6 production. All cytokine measurements were performed in triplicate.

### 2.7. Quantification by Reversed-Phase HPLC-DAD Analysis

Analyses of CE compositions were performed by liquid chromatography using apparatus equipped with a quaternary pump, diode array detector, Varian 445 LC-DAD column compartment temperature control on an ACE^®^ 5 C18-column (250 × 4.6 mm, 5 Å). The used solvents were (A) water–acetonitrile–formic acid (98, 0.8, 0.2, *v*/*v*/*v*) and (B) acetonitrile–formic acid (98.08, 0.2, *v*/*v*) (Carlo Erba (Milan, Italy)). The analysis was made with the following gradient: 0.1–4 min for 5%B, 4–15 for 5%B, 15–75 for 13%B, 75–85 for 35%B, 85–95 for 100%B, 95–95.5 for 100%B, and 95.5–105 for 5%B. Flowing at 0.7 mL/min, 10 μL of CE was injected at 10 g/L and analysis was conducted with a DAD detector at 280 nm. Results of the quantification of theobromine (Alfa Aesar, Ward Hill, MA, USA), caffeine (Alfa Aesar, Ward Hill, MA, USA), epicatechin, and procyanidins A1, A2, B2 and C1 (Extrasynthese, Lyon, France) were expressed in mg of compounds per 100 g of dry matter (mg/100 g DM). All quantifications were performed in triplicate.

### 2.8. Statistical Analysis

Statistical significance was determined using non-parametric analysis (Mann–Whitney U test) using STATA\IC-version 12. Significance was accepted at the 5% level (*p* ≤ 0.05).

## 3. Results

### 3.1. Extractions

Due to possible interferences of carotenoids and apolar compounds such as lipids in AO assay measurements [23], defatting of cocoa powders before polyphenols extraction was necessary. A significant lipid level difference was noticed in both varieties. Indeed, unfermented Guiana (31.15 ± 0.13%) would have less fat than unfermented Forastero (34.85 ± 2.61%). After six days of fermentation, although varietal difference persisted, no significant impact on lipid rates was noticed for Guiana (24.65 ± 1.30%) and Forastero (37.44 ± 1.65%) fermented beans. Regarding the extraction with the binary solvent (acidified water–acetone, 30/70, *v*/*v*), fermentation induced a significant impact. After six days of fermentation, extraction yield increased in 1.9% and 2.1% in Guiana and Forastero varietals, respectively. No varietal difference was noted whatever the process used.

### 3.2. Antioxidant Activity

All results dealing with TPCs, TFCs and antioxidant activities are shown in Table 1.

If no varietal differences were noticed in the TFC assays (308.64 ± 10.1 and 306.5 ± 4.7 μg CE/g DM for unfermented Guiana and Forastero, respectively), a significant varietal difference was noticed in TPCs (41.3 ± 0.8 and 45.9 ± 1.2 mg EAG/g DM for Guiana and Forastero, respectively).

In vitro AO activities cannot be determined with a single method; therefore, we have selected a few with various AO mechanisms to do so. FRAP induces an electron transfer, while DPPH induces a radical electron transfer and ORAC induces a proton-transfer mechanism. Unfermented Guiana presented equivalent DPPH activity (294.9 ± 17.9 μmol ET/g DM) and FRAP activity (684.6 ± 48.5 μmol Fe_2_^+^/g DM) with unfermented Forastero. However, unfermented Guiana presented a 1.3-fold higher activity (1097 ± 111.8 μM ET/g DM) than unfermented Forastero (838.5 ± 67.8 μM ET/g DM) in the ORAC assay.

Furthermore, we noticed that fermentation induced a significant decrease in TPCs, TFCs and AO activities for both varieties. Indeed, after six days of processing, TPCs decreased by 32% in Guiana and 20% in Forastero. Additionally, Guiana showed a decrease in 25, 31 and 23% for its DPPH, FRAP, and ORAC activities, respectively, after six days of processing.

### 3.3. Polyphenolics and Methylxanthins Quantification

Chromatograms of both varieties are presented in Figure 1.

The chromatographic profiles are quite similar between Guiana and Forastero, which might indicate that they might be qualitatively similar; 10 compounds, some of which were majority contents, were found. However, some quantifiable differences could be noticed, and all quantifications of cocoa compounds are presented in Table 2. Cacao is a major source of polyphenols and methylxanthines, particularly epicatechin and theobromine, which could be responsible for numerous AO and AI activities. In this work, Guiana showed epicatechin contents of 614.1 ± 39.3 mg/100 g DM and theobromine contents of 1180.0 ± 33.0 mg/100 g DM.

Both varieties underwent a major decrease in their compound contents after six days of fermentation. For example, the procyanidin A1 levels decreased by 58.2% in Guiana and by 35.8% in Forastero.

### 3.4. Immunomodulatory Activities

#### 3.4.1. Viability of J774A1 Macrophages

Before studying the impact of extracts on the inflammation process on macrophages, CEs were tested to confirm their non-toxicity with a viability assay using MTS and PMS. After 24 h in the presence of CEs, results confirmed the non-cytotoxic impact on macrophages for both varieties (Figure 2).

#### 3.4.2. Quantification of Inflammatory Markers

In our study, we selected a pre-treatment method of 4 and 24 h. This choice was oriented by the fact that before 4 h of pre-treatment, cytokine contents were too low to be measured. Additionally, with a pre-treatment time greater than 24 h, macrophages could produce toxics products (such as cytokines), that could cause their death. We pre-treated macrophages with Guiana and Forastero CEs with only unfermented samples those and fermented for two and six days. These fermented samples were chosen because they represented the two stages of the fermentation process: alcoholic and acetic fermentation, respectively. The three fermentation times were analyzed at three concentrations: 12.5, 25 and 50 μg/mL.

LPS/IFNγ stimulation is well known to cause a significant production of NO, TNF-α or IL-6, by inducing inflammation in macrophages. The impact of CE on this production could be determined with the ratio of cytokines produced with or without CE before LPS/IFNγ stimulation.

In the case of NO production in our study, no impact was observed with a 24 h pre-treatment of CE at different concentrations. Moreover, fermentation might not modulate NO production either; the slightly stimulating rate, which was noticed for the CE fermented for six days, was not significant to confirm an impact. This trend was noticed for both varieties in Figure 3.

Conversely, a stimulation of the TNF-α production was noticed after a 24 h pre-treatment with the Guiana CEs at all concentrations (Figure 4). Indeed, a concentration-dependent effect was noticed, with maximum activity at 50 μg/mL. Guiana CEs presented a stimulation rate of 111.9 ± 2.6% (unfermented), 57.76 ± 3.4% (fermented for two days) and 196.63 ± 30.7% (fermented for days). This trend was observed with Forastero CEs (unfermented and fermented for six days). At 50 μg/mL, Forastero CEs demonstrated a TNF-α stimulation by 132.7 ± 6.7% (unfermented), 67.76 ± 11% (fermented for two days) and 124.47 ± 18.6% (fermented for six days). No varietal differences were noticed. Nevertheless, fermentation seemed to impact this action by increasing the stimulating capacity of extracts after fermentation for six days.

As described below, LPS/IFNγ stimulation caused a significant accumulation of IL-6 in the culture medium. Unlike TNF-α, CE pre-treatment at different concentrations significantly inhibited the LPS/IFNγ-induced IL-6 production (Figure 5). Indeed, with a maximum at 50 μg/mL, we observed an impact of CE in a concentration-dependent manner after a pre-treatment of 4 h. At 50 μg/mL, Guiana CEs showed an IL-6 inhibition rate of 56.8 ± 0.9% (unfermented), 50.2 ± 2.5% (fermented for two days) and 49 ± 0.1% (fermented for six days). Guiana demonstrated a better inhibition than Forastero, which showed inhibition rates by 45.5 ± 0.2% (unfermented), 40.8 ± 2.3% (fermented for two days), and 21.9 ± 0.7% (fermented for six days).

This inhibition appeared after 4 h of pre-treatment, whereas no actions in TNF-α at this same incubation time were noticed (Appendix A). Furthermore, no actions were noticed at 24 h for IL-6 production, whereas stimulation of TNF-α was recorded (data not shown). Consequently, CEs indicated pro-inflammatory capacities by activating TNF-α production after 24 h of pre-treatment, whereas CEs presented an anti-inflammatory by inhibiting IL-6 production after 4 h of pre-treatment. 

## 4. Discussion

### 4.1. Extractions

A varietal difference was noted between Guiana and Forastero, despite a very small difference in their lipid rates. This characteristic was also noticed in another comparative study involving Guiana (52.1%) and Forastero-type cocoa “Amelonado” (53.2%) from French Guiana [24]. In our study, for both varieties, lipid levels were lower (20%) than those found in the literature. Indeed, Servent et al. (2018) found lipid rates between 50 and 56% for Trinitario and Criollo varieties, respectively [25]. Lower rates in our study may be explained by the choice of lipid extraction procedure which was used. Maceration and ultrasonic extractions were preferred as gentler procedures, rather than Soxtec and Soxhlet extractions at 100–110 °C to avoid the degradation of thermosensitive compounds, such as polyphenols. After six days of fermentation, the analysis of lipid rates showed that no significant impact of fermentation was noticed either both varieties. Fermentation already demonstrated that no changes could be observed in lipid content during this process [25].

Extracted with ethanol (80% *v*/*v*), an Indonesian cocoa has demonstrated an extraction yield of 12.95% [26]. According to the literature, a mix of water–acetone is appropriate for the extraction of high molecular weight polyphenols, such as proanthocyanidins and ellagitanins [27,28]. Indeed, acetone allows a better extraction of anthocyanins and flavonols, with intermediate efficiency for flavan-3-ols, procyanidins and ellagic acid derivatives. However, using only this solvent is less effective for hydroxycinnamic acid derivative extractions. The addition of formic acid could give a slightly better efficiency for these compounds and much better efficiency for flavan-3-ols and ellagic acid derivatives [27]. Guiana has showed an equivalent extraction yield than Forastero and a qualitative and quantitative study on Guiana composition was necessary to deepen these data. A significant impact of fermentation on extraction yield was noticed for both varieties (Guiana 18.03 ± 0.6 to 20.7 ± 1.1%; Forastero 18.13 ± 1.0 to 20.36 ± 0.5%). This difference before and after fermentation could be due to internal changes in the bean which create more accessibility to the target-compounds and allow a better extraction by the solvent. Indeed, cocoa cells are composed by 90% of uncolored cocoa cells (which contained sugars, proteins, enzymes and 50% of lipids), and 10% colored cells (polyphenols and methylxanthines). All of them are covered with a lipid film which isolates them from each other. After fermentation, acetic acid converts the lipid film into lipidic globules and allows substrates and enzymes to meet. Proteins are converted into free amino acids, complex sugars become simple sugars and may complex with polyphenols, which could also be converted into insoluble polymers [29]. The increase in extraction yield could be due to the fact that at 6 days, more compounds were accessible to the solvent and the complexation of polyphenols and free sugar and/or free amino acids could induce the extraction of bigger compounds.

### 4.2. Antioxidative Activity

By comparing our results with those of the literature we observed that Guiana has an equivalent DPPH activity to two blends of Brazilian cocoa hybrids which have AO capacities of 324.06 ± 51.7 and 363.3 ± 120.1 μmol TE/g DM [30,31]. For ORAC assay, Guiana revealed a better AO capacity than Africa Forastero (303 ± 5 μmol TE/g DM) [12], Colombian Forastero (540.2 ± 51.2 μmol TE/g DM) and a blend of Colombian Trinitarios (507.2 ± 78.86 μmol TE/g DM) [32]. However, Guiana was less effective than some Brazilian Forastero cocoa hybrids with an average of 2059.7 ± 565.2 μmol TE/g DM [30]. Compared with two Forastero hybrids that have FRAP capacities of 822.1 and 795.9 μmol Fe^2+^/g DM [33], Guiana (with an average of 645.3 ± 0.8 μmol Fe^2+^/g DM) has an equivalent capacity.

Commonly compared to another widely consumed daily food, unfermented Robusta green coffee (*Caffea canephora*) demonstrated a DPPH capacity of 116 ± 15 μmol TE/g DM [34], while Arabica green coffee (*Caffea arabica*) demonstrated DPPH capacities of 83 ± 18 and 1627.09 ± 158.7 μmol TE/g DM) [34,35]. Others coffee species such as *Caffea benghalensis* and *Caffea Liberica* also demonstrated high DPPH activities of 1691.4 ± 153.3 and 2212.8 ± 204.1 μmol TE/g DM, respectively [35]. In ORAC assay, Arabica and Robusta coffees showed respectively a two-fold higher AO capacity (with 1821 ± 345 μmol TE/g DM) and 2.6-fold higher (with 2594 ± 71 μmol TE/g DM) than Guiana [36]. Compared to unfermented green beans coffee, Arabica and Robusta have demonstrated very similar FRAP capacities (553 ± 73 and 692 ± 31 μmol Fe^2+^/g DM, respectively) to Guiana [34]. Consequently, Guiana can be considered as equivalent to some coffee varieties in FRAP assays, but less antioxidant in ORAC and DPPH assays.

No varietal difference was observed in FRAP and DPPH assays, but Guiana has a higher ORAC capacity than Forastero. DPPH and ORAC are both based on antiradical mechanisms, but the DPPH• radical is a non-natural radical and cannot be found in organisms. In contrast, ORAC is based on AAPH use which, in the presence of oxygen, leads to the formation of peroxyl radicals ROO•. These radicals are naturally produced in living biological organisms. Consequently, Guiana demonstrated a better action on a more realistic test, which may indicate that Guiana is more effective than Forastero. The difference in the AO capacities for both cocoa varieties could be explained by a different composition, because the correlation between AO capacity and polyphenols compounds has already been demonstrated. For example, Todorovic et al. (2015) showed that cocoa powder, which presented a three-fold higher TPC than white chocolate, has 17-fold and 11-fold higher DPPH and ORAC activities, respectively [37]. In our case, the Guiana TPC was 0.9-fold lower than Forastero and demonstrated a more efficient ORAC activity. Knowing that TPC assay quantifies a set of polyphenols, Guiana’s different ORAC capacity could be linked to its own quantitative and qualitative distribution in terms of compounds.

Fermentation induces a degradation of compounds within cocoa beans, particularly polyphenols. Indeed, polyphenol oxidase and glycosidase would be responsible for the conversion of polyphenols into quinones, which will complex with other cellular compounds. Additionally, the acidic conditions of the medium induce hydrolysis of polyphenols [38]. With an initial concentration by 73.48 GAE/g DM, Colombian hybrid CCN-51 sustained a decrease, with values at two, four, and six days of fermentation of 69, 47.85 to 36.68 mg GAE/g DM, respectively, which represented a total loss of 51% [39]. In the same way, in our study, TPCs reduced by 32% and 20% for Guiana and Forastero, respectively. Consequently, if the phenol composition decreased during this process, fermentation would also contribute to a decrease in AO activities. The analysis of correlation between TPCs and AO activities (Appendix A) showed an *R*^2^ value of 0.98 for FRAP, 0.89 for DPPH, and 0.81 for ORAC, suggesting that a good correlation exists between total phenolic contents and AO capacities. In the literature, this correlation was observed after six days of fermentation: Forastero hybrids TSH565 and ICS60 TPCs decreased by 35% and 20%, while DPPH activity decreased also by 25% and 8%, respectively [30]. Furthermore, differences between Guiana and Forastero may also result from differences in polyphenol composition that are not reflected in TPCs and TFCs.

### 4.3. Polyphenolics and Methylxanthins Quantification

Analysis of our chromatograms showed that we identified the main compounds commonly found in cacao species. Indeed, methylxanthines and phenols as procyanidins and epicatechins have been found [40]. Methylxanthine contents might be variable in cocoa seeds. Indeed, Febrianto and Zhu found theobromine contents between 1940 and 3170 mg/100 g of DM, and caffeine contents between 53 and 554 mg/100 g of DM [41]. These compounds participate in the aromatic bouquet of cocoa [41]. Guiana turned out to be significantly (15%) richer in theobromine and had 50% less caffeine than Forastero. This methylxanthine content could be characteristic of the variety Guiana, used as a varietal marker and being responsible for its unique flavor. Indeed, for a panel of 15 qualified members, chocolate made with Guiana has presented a higher overall aroma intensity and cocoa flavor than Forastero-type Amelonado [24]. Criollo, which is very appreciated for its fine aroma, is hardly cultivable because it is susceptible to diseases [42]. Guiana, with its agronomical performances [43] and characteristic flavors, could be a good alternate strategy.

No significant differences were identified in the epicatechin and procyanidins contents between Guiana and Forastero. As for methylxanthines, polyphenol contents could be variable, depending on factors such as variety or geographical origin. Indeed, for Forasteros from Nigeria, Cameroon, Ivory Coast, Ghana, and Ecuador, epicatechin contents were between 151 ± 6 and 451 ± 20 mg/100 g DM and procyanidin B2 were between 68.4 ± 15 and 157 ± 16 mg/100 g DM [44]. A significant difference in TPC and equivalent epicatechin and procyanidin A1, A2, B2 and C1 contents between both varieties suggest that Guiana and Forastero could be composed of a set of various phenolic compounds that might be in varying amounts.

One of the aims of fermentation is the reduction in bitterness of the bean; this could be possible due to the external diffusion of methylxanthines (30%) and polyphenols (20%), caused by the increase in permeability of cells of the seed [45]. In our study, after six days of fermentation, caffeine decreased by 27% in Guiana and 36% in Forastero, whereas no significant decrease in theobromine level was noticed for either variety. This might be due to the diffusion of theobromine from the almond to the shell. Indeed, Hernández-Hernández et al. (2018) have demonstrated that, during fermentation, the initial and final theobromine contents of the bean (almond and shell) were equivalent, but the distribution of theobromine was different. For example, the theobromine of almond decreased by 1807 ± 1 to 979 ± 3 mg/100 g DM while those of shell increased by 390 ± 15 to 1200 ± 10 mg/100 g DM, after six days [40]. The small difference between initial and final theobromine contents in Guiana may be due to the fact that we kept the shell of our beans in our analyses. Added phenomena to this diffusion process are enzymatic degradations and acidic hydrolysis, which affected the polyphenol content. We noticed a reduction in epicatechin by 46% in Guiana and 60% in Forastero after six days of fermentation. Under the same conditions, Guiana’s procyanidin contents decreased by 39% for A2, 49% for B2, and by more than 54% for A1 and C1. In the literature, losses of epicatechin by 40 to 92% and procyanidin B2 by more than 70% were observed for Forastero-type cocoa [30,46].

With epicatechin contents of 614 ± 0.39 mg/g DM and procyanidin contents (dimer and trimer) of 512 ± 0.21 mg/g DM, the valorization of Guiana into chocolate could be a good alternative to obtain new sources of antioxidants. For Guiana cocoa beans, after six days of fermentation, we observed a loss of 46% and 49% for epicatechin and procyanidins, respectively. Bordiga et al. (2015) have demonstrated that the transformation of fermented beans to chocolate induced an average loss of epicatechin by 81.8 ± 8.1% for African cocoa [44]. Consequently, if Guiana fermented for six days was turned into chocolate, we would obtain final epicatechin contents of 1.116 ± 0.071 mg/g DM. Guiana chocolate epicatechin content could be 1.4-fold and 2.4-fold higher than two other dark chocolate contents [44]. The fact that dark chocolates have also demonstrated AO activities in ORAC assay (227 ± 74 μM TE/g DM) [44] suggests that Guiana chocolate could be a good cocoa-derivate with AO capacities.

### 4.4. Immunomodulatory Activities

In the literature, polyphenols seemed to be responsible for AI activities in macrophages by modulating key inflammation-regulators such as TNF-α, IL-1-β and IL-6 in vitro and in vivo [15]. According to some studies, cocoa has showed inhibitor activities in NO production. For example, CE has demonstrated inhibition rates from 17 to 72.3% at 5–100 μg/mL [47]. Although the fact that this experiment operated in this concentration range, we did not notice any significant impact on NO production. That could be explained with differences in extraction procedures (water and ethanol extraction), which could engender qualitative and quantitative differences in the composition of the two CEs. Another hypothesis would be the weak concentration of epicatechin in our CE. Indeed, Ramiro et al. (2005) have demonstrated that epicatechin at final concentrations of 200 and 400 μM inhibited NO production by 32.3 and 49.2%, respectively [47]. However, in our study, at our maximal concentration (50 μg/mL), we used epicatechin at 5.86 × 10^−6^ μM for unfermented Guiana and 6.11 × 10^−6^ μM for unfermented Forastero. These small amounts of epicatechin in our analysis could explain the absence of NO production modulation. Moreover, we used a pre-treatment of 24 h, whereas Ramiro et al. (2005) used a simultaneous action of CE and epicatechin with inflammatory stimuli (added at the same time).

Even though most of the time cocoa could inhibit inflammatory cytokines such as TNF-α, some authors have reported the stimulating capacities of TNF-α production, which would confirm that cocoa has a pro-inflammatory potential. For example, CEs which were incubated for 72 h in another inflammatory cell model (peripheral blood mononuclear cell, PBMC) have demonstrated a TNF-α production stimulation after the initiation of inflammatory process by phytohemagglutinin (PHA). Indeed, in stimulated PBMC, stimulation rates of TNF-α production were noticed, with a maximum of 128% for hexamers and a minimum of 1.5% for monomers (epicatechin and catechin, respectively). Moreover, in non-stimulated PBMC, CE also induced a stimulation in TNF-α production by 42–183% with phenolic fractions (from monomers to decamers, respectively) with a maximum of 412% for hexamers [48]. Consequently, procyanidins, which are presented in large amounts in cocoa, might be responsible for TNF-α stimulation capacities.

In our study, we determined the procyanidin contents (dimers and trimers) in both unfermented cocoa samples at 50 μg/mL. Guiana presented a maximum of 0.7 μg/mL of dimer B2 and 0.5 μg/mL of trimer C1, whereas impacts were noticed at 25 μg/mL of fraction cocoa polymers in the study of Mao et al. (2002). We used a 24 h pre-treatment to determine the impact of CEs on TNF-α production, whereas Mao et al. started the inflammatory stimulation at the same time as the CE treatment. Other authors have also applied pre-treatments instead of simultaneous treatments and demonstrated stimulating effects. For example, cells pre-treated with short-chain flavonoid fractions (SCFFs—monomers to pentamers) and those with long-chain flavonoid fractions (LCFFs—pentamers to decamers) for 16 h before stimulation, demonstrated TNF-α stimulations of 36% and 90%, respectively [49].

Moreover, fermentation, which was demonstrated to diminish the polyphenol content, particularly epicatechin and procyanidins, seemed to improve the stimulating capacity on TNF-α production. This trend could reinforce the hypothesis that polyphenols are not the only responsible for this potential. Otherwise, TNF-α, which is often illustrated as a negative cytokine, could play a role in immunocompromised metabolisms. TNF-α demonstrated an antitumoral potential by inducing the apoptosis of dangerous cells, but also by inducing the resolution of inflammation through triggering cell recruitment [50,51].

Unlike the previous two actions, Guiana has shown an anti-inflammatory action on the IL-6 production. This anti-inflammatory capacity has already been found in other studies. Indeed, with human monocytes pre-treated for 1 h with CEs, and then stimulated, Zeng et al. (2011) noticed an inhibition of IL-6 production by 24 ± 5% (0.1 μg/mL) to 65 ± 7% (10 μg/mL) [52]. As for the NO inhibition, epicatechins seemed to be responsible for cocoa’s anti-inflammatory properties. Indeed, a 24 h pre-treatment with epicatechin at 5, 25 and 50 μM induced inhibitions of 13, 25.5 and 36% of IL-6 production, respectively, for RAW macrophages stimulated with LPS [53]. In our study, at 50 μg/mL, Guiana has shown an inhibition rate of 56.8 ± 0.9% with an epicatechin content of 5.86 × 10^−6^ μM for an unfermented sample and 49% ± 0.1 for 2.7 × 10^−6^ μM for a sample fermented for six days. These appear to be enough to induce an inhibition of IL-6 production, although the possibility of a cocktail effect with other compounds should not be overlooked.

Consequently, in our CE, procyanidins seemed to be responsible for the TNF-α production stimulation, and monomers such as epicatechin seemed to be responsible for IL-6 production inhibition. The procedure seemed to also have an impact, and various actions could depend on the fact that CE was used before or simultaneously with the pro-inflammatory stimuli. These opposite immunomodulatory capacities could be due to the balanced relationship between IL-6 and TNF-α [54].

Indeed, an IL-6 deficit contributes to a suppression of high TNF-α levels in infected mice by *Streptococcus pyogenes* [55]. This could explain the absence of TNF-α production at 4 h. Then, at 24 h, a loss of IL-6, might induce a TNF-α production, which was able to self-amplify itself and might cause a TNF-α overproduction [51]. This modulation could be under the control of NF-κB, a nuclear factor which might play a role in DNA transcription and cytokine production. In the literature, dietary polyphenols have showed modulations in NF-κB activation and reductions in inflammation process [56]. Additional studies will be useful to improve our knowledge about the immunomodulatory action of Guiana CEs.

## 5. Conclusions

To the best of our knowledge, this is the first time that Guiana biochemistry and health benefits have been characterized. The few existing studies involving this variety only focused on its agronomic performances or genetic distinctions with other varieties. Our work highlighted the in vitro AO capacity of Guiana and showed that it would be as efficient as other cocoa varieties. Polyphenols were correlated with these AO activities. Guiana presented a similar phenol content (mainly epicatechins or procyanidins) when compared to Forastero. A varietal difference could explain methylxanthine content differences, because Guiana contains less caffeine and more theobromine than Forastero. This characteristic, leading to a lower astringency, could induce a fine aroma and a reduction in the fermentation process time. Moreover, because fermentation is also responsible for a 30% decrease in AO potential, this reduction in process time could potentially maintain the benefits of Guiana, while having its characteristic flavors.

Furthermore, Guiana has demonstrated a better inhibition in IL-6 production than Forastero, but also a stimulation in TNF-α production at two pre-treatment times. This is the first time that Guiana has been demonstrated to show opposite actions on inflammatory assays. This behavior shows that cocoa is a complex matrix which presents different attributes depending on how it is used (preventive or curative aims). Fermentation could modulate this immunomodulatory capacity by improving the TNF-α stimulation of Guiana CEs. Indeed, further studies might be conducted to allow a better comprehension of these AO and AI capacities. Moreover, study using fractions of compounds could be a good strategy to study bioguiding and synergic effects.

## Figures and Tables

**Figure 1 foods-10-00522-f001:**
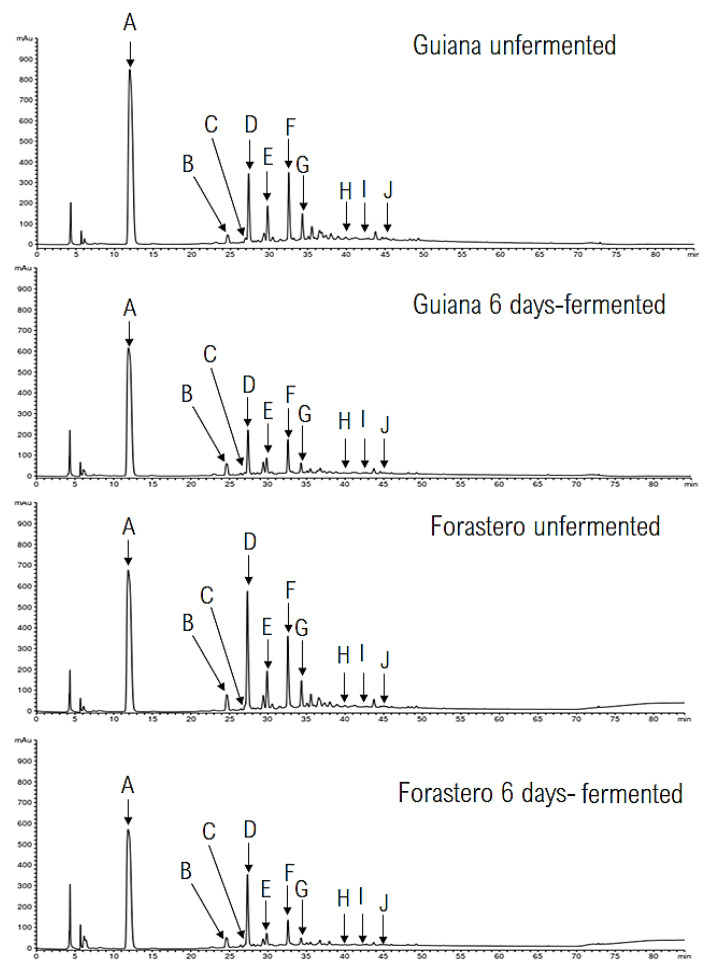
Liquid chromatography chromatograms of Guiana and Forastero 2018 extracts (unfermented and fermented for six days). A: Theobromine, B: Procyanidin B3 + unknown compound, C: Catechin, D: Caffeine, E: Procyanidin B2, F: Epicatechin, G: Procyanidin C1, H: Procyanidin A1, I: Epicatechin gallate, J: Procyanidin A2.

**Figure 2 foods-10-00522-f002:**
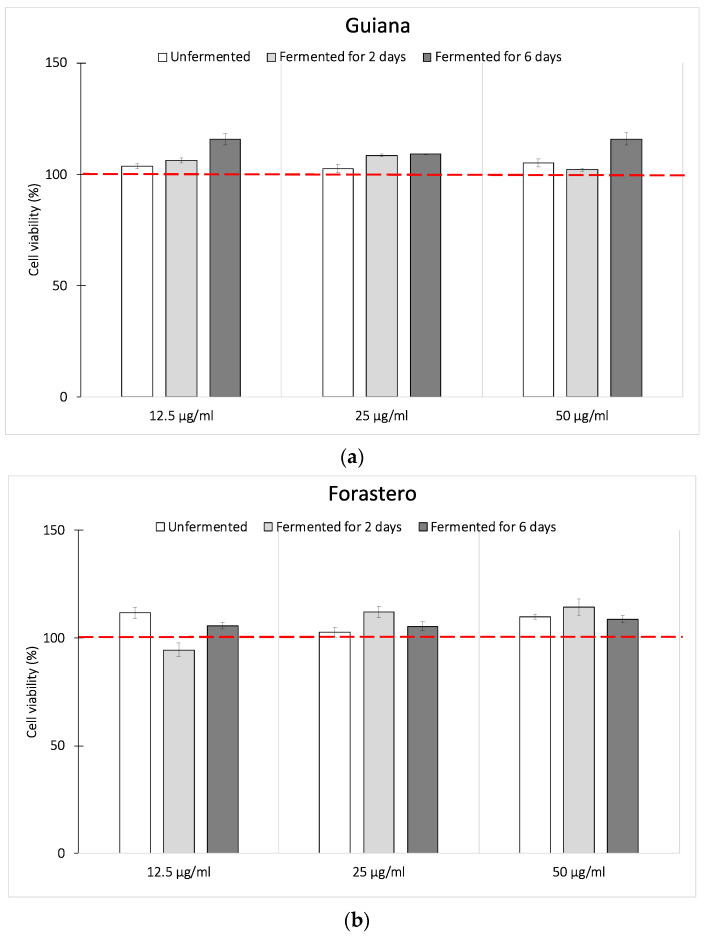
Cell viability of J774A1 macrophages pre-treated for 24 h with various concentrations of Cocoa extract (CE). The cell viability was determined by the 3-(4,5-dimethylthiazol-253 2-yl)-5-(3-carboxymethoxyphenyl)-2-(4-sulfophenyl)-2H-tetrazolium (MTS) and phenazine methosulfate (PMS) assay. (**a**) Guiana, (**b**) Forastero. Results represent the mean viability rate ± SEM (*n* = 5).

**Figure 3 foods-10-00522-f003:**
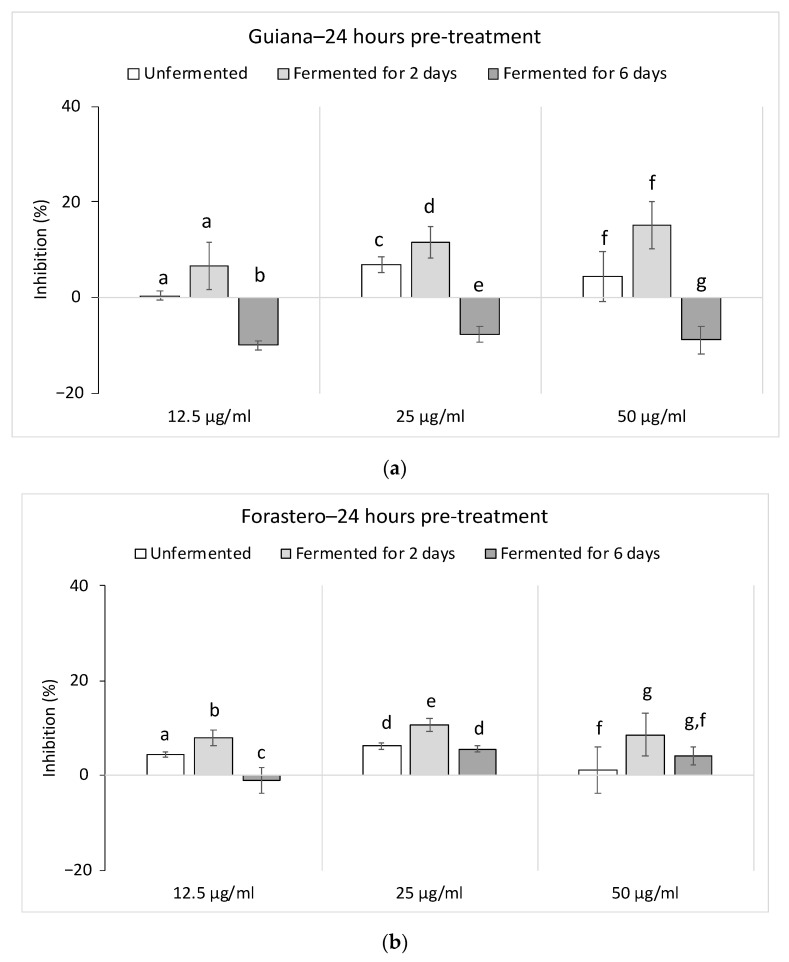
Inhibition rates of both varieties on nitric oxide (NO) production in lipopolysaccharide (LPS) and interferon gamma (IFNγ)-stimulated macrophages. (**a**) Guiana, (**b**) Forastero. Results represent the mean % inhibition of NO-production ± SEM (*n* = 4). At each concentration, histograms with the same letter ^a–g^ for (**a**) or (**b**) were not significantly different (*p* < 0.05) using a Mann–Whitney U test.

**Figure 4 foods-10-00522-f004:**
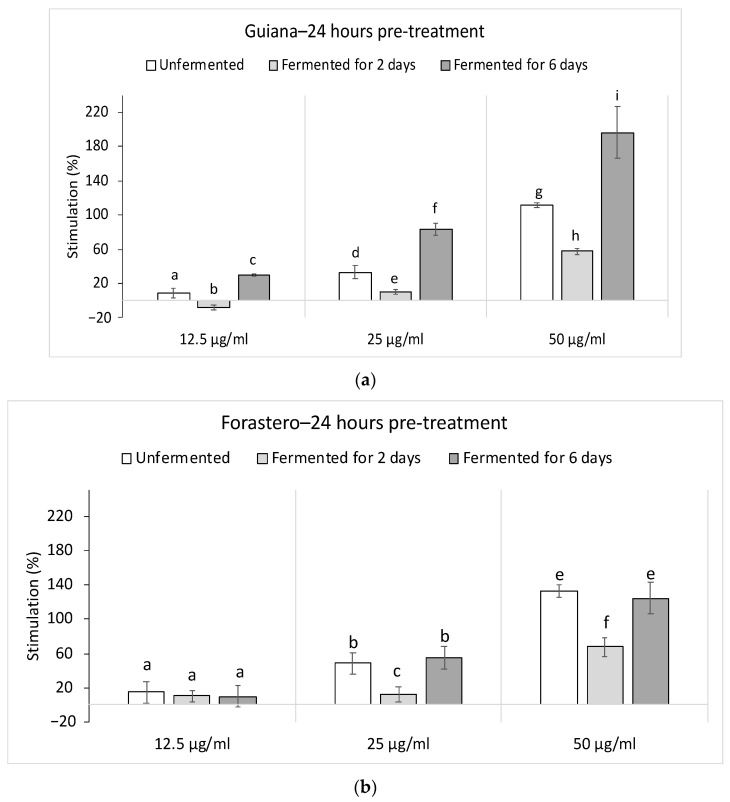
Stimulation rates of both varieties on tumor necrosis factor alpha (TNF-α) production in lipopolysaccharide (LPS) and interferon gamma (IFNγ)-stimulated macrophages. (**a**) Guiana, (**b**) Forastero. Results represent the mean % stimulation of TNF-α production ± SEM (*n* = 4). At each concentration, histograms with the same letter ^a–i^ for (**a**) or (**b**) were not significantly different (*p* < 0.05) using a Mann–Whitney U test.

**Figure 5 foods-10-00522-f005:**
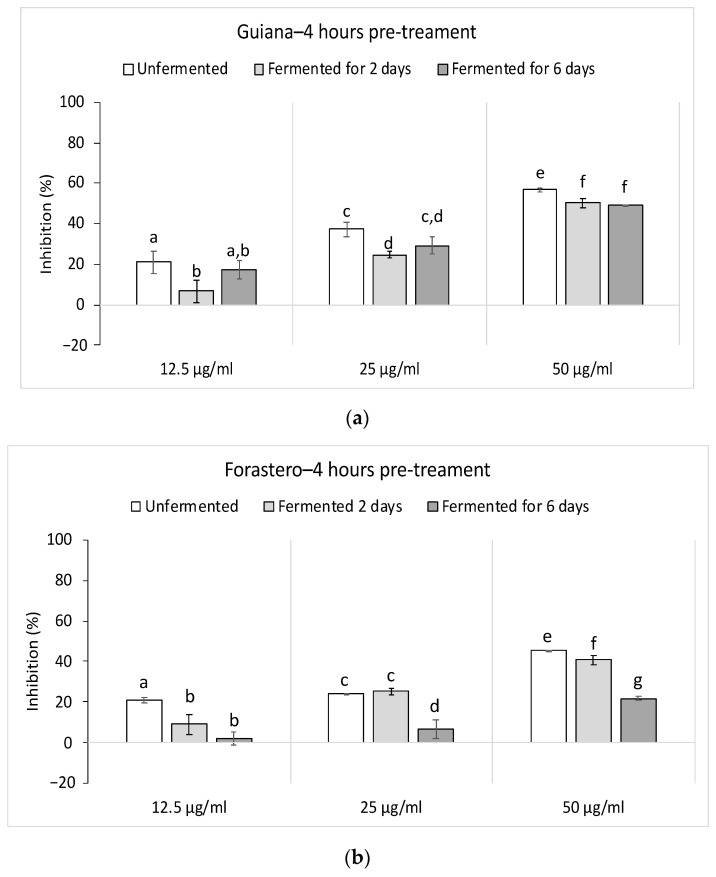
Inhibition rates of both varieties on interleukin-6 production in lipopolysaccharide (LPS) and interferon gamma (IFNγ)-stimulated macrophages. (**a**) Guiana, (**b**) Forastero. Results represent the mean % inhibition of IL-6 production ± SEM (*n* = 4). At each concentration, histograms with the same letter ^a–g^ for (**a**) or (**b**) were not significantly different (*p* < 0.05) using a Mann–Whitney U test.

**Table 1 foods-10-00522-t001:** Guiana and Forastero antioxidant activities, total polyphenolic and total flavonoid contents.

			TPC (mg GAE/g DM)	TFC (μg CE/g DM)	DPPH (μmol TE/g DM)	FRAP (μmol Fe^2+^/g DM)	ORAC (μM TE/g DM)
Guiana	Unfermented		41 ± 0.8 ^a^	309 ± 10.1 ^a,c^	295 ± 17.9 ^a,c^	685 ± 48.5 ^a,d^	1097 ± 111.8 ^a^
	Fermented (days)	2	35 ± 0.9 ^b,f^	290 ± 10.6 ^a,d^	247 ± 10.5 ^b^	613 ± 44.6 ^a^	1059 ± 54.9 ^a^
		4	38 ± 3.8 ^a,b,g^	312 ± 18.0 ^a^	256 ± 16.5 ^b^	534 ± 28.5 ^b^	983 ± 97.3 ^a,b^
		6	28 ± 1.7 ^c^	148 ± 7.2 ^b,d,g^	221 ± 17.5 ^b,e^	450 ± 25.6 ^c^	848 ± 48.3 ^b^
Forastero	Unfermented		46 ± 1.2 ^d^	307 ± 4.7 ^c,e^	329 ± 42.4 ^c,d^	827 ± 73.1 ^d,e^	839 ± 67.8 ^c,d^
	Fermented (days)	2	52 ± 2.1 ^e^	329 ± 8.5 ^e^	351 ± 2.5 ^d^	987 ± 26.6 ^c^	1133 ± 22.5 ^e^
		4	42 ± 1.5 ^a^	233 ± 16.5 ^f^	293 ± 17.1 ^d^	754 ± 58 ^e^	860 ± 93.4 ^d^
		6	37 ± 2.0 ^f,g^	133 ± 4.2 ^g^	247 ± 22.7 ^e^	575 ± 6.8 ^f^	696 ± 43.9 ^f^

Averages with the same letter ^a–g^ within columns were not significantly different (*p* < 0.05) using a Mann–Whitney U test. TPC, total polyphenol content; TFC, total flavonoid content; GAE, gallic acid equivalent; TE, Trolox equivalent; DM, dry matter; DPPH, 2,2-diphenyl-1-picrylhydrazyl; FRAP, ferric reducing antioxidant power; ORAC, oxygen radical absorbance capacity. The results are expressed as the mean ± standard deviation (*n* = 9).

**Table 2 foods-10-00522-t002:** Influence of variety and fermentation in cocoa chemical compositions.

Compound	Guiana		Forastero	
	Unfermented (mg/100 g DM)	Fermented for 6 days (mg/100 g DM)	Unfermented (mg/100 g DM)	Fermented for 6 days (mg/100 g DM)
Theobromine	1180 ± 33.0 ^a,1^	968 ± 207.8 ^a^	1005 ± 84.3 ^b,2^	921 ± 51.7 ^b^
Caffeine	160 ± 13.7 ^a,1^	116 ± 14.9 ^b^	315 ± 44.9 ^c,2^	204 ± 23.0 ^d^
Epicatechin	614 ± 39.3 ^a,1^	332 ± 29.0 ^b^	641 ± 66.1 ^c,1^	257 ± 9.3 ^d^
Procyanidin A1	36 ± 4.0 ^a,1^	15 ± 0.7 ^b^	20 ± 2.4 ^c,1^	13 ± 1.9 ^d^
Procyanidin A2	39 ± 3.1 ^a,1^	24 ± 2.8 ^b^	32 ± 4.9 ^c,1^	28 ± 1.7 ^c^
Procyanidin B2	254 ± 14.8 ^a,1^	130 ± 20.7 ^b^	263 ± 17.7 ^c,1^	125 ± 12.6 ^d^
Procyanidin C1	178 ± 23.5 ^a,1^	82 ± 2.9 ^b^	182 ± 25.8 ^c,1^	78 ± 11.4 ^d^

Averages with same letter a–d within lines were not significantly different (*p* < 0.05) using a Mann–Whitney U test. For unfermented samples, the same number 1–2 within rows indicates that no varietal difference was noticed (*p* < 0.05) using a Mann–Whitney U test. Results are expressed as the mean ± standard deviation (*n* = 9).

## Data Availability

The data presented in this study are available on request from the corresponding author. The data are not publicly available due to it is not in an online archive.

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
