# Peer review of "Antioxidative and Immunomodulatory Potential of the Endemic French Guiana Wild Cocoa “Guiana”"

_foods, 2021, doi:10.3390/foods10030522_

Round 1
Reviewer 1 Report
This paper deals with the evaluation of the antioxidant and immunomodulatory capacities of cocoa Guiana and a comparison with the classic Forastero variety. Moreover, the content of phenolic compounds and methylxanthines were determined by HPLC-DAD.
This is the first study on the chemical characterization of this variety of cocoa as well as the bioactivities due to the bioactive compounds. The study was well conducted and may attract the attention of cocoa experts and the scientific community.
I recommended the publication of this study in the Journal Foods after the following minor revisions:
1- Abstract: results related to the bioactive compounds determined by HPLC responsible for the bioactivities should be included.
2- Lines 213-215: The sentence should be rewritten because is intricate.
3- Lines 219-228: The authors should indicate that this data is for unfermented cocoa beans.
4- Table 1: I suggest to authors to change the presentation of the results and change the lines by the columns. In this way, the statistical results will be easier to read since will be in the same columns instead of in two blocs of data as presented.
5- Still for table 1: the authors should be correct the significant numbers of the results. Seams that no criteria have been taken.
6- Table 2: The authors have some explanation for the fact for the lower impact of the fermentation process on the decrease of active compounds? some other components? The chemical or biological reason?
7- Table 2: Some differences in the content concerning variety can be lost with the statistical analysis performed since the fermentation process has a high impact on the total contents. However, if the authors compare raw beans some differences could be noticed that are related to the variety.
8- Al figures: The numbers separator on the graphics should be replaced by a dot (.)
9- Lines 299-308 and Fig. 4: Since the active compounds are lower? the concentration should be adjusted? Since for Forastero, the differences were not noticed?
The author should include statistical data in these figures.
10- Lines 327-329: If the authors agree, these data could be included as supplementary data.
11- Lines 377-387: I suggest to authors eliminate this part of the coffee since data on cocoa beans are available in the literature for discussion, otherwise include other foods also common in the diet.
12- Line 398: "In our case, the TPC of Guiana is different from Forastero." How different? The authors should complete the sentence. In your case is also different, how much x-fold...
13- Lines 488-494: Theobromine was not considered as a potential bioactive compound, it could be? Some recent studies have demonstrated some anti-inflammatory activity of theobromine. Maybe in the future, the authors should test also authentic standards in order to verify the molecules responsible for the activity.
14- Section 3.4.: Some discussion on anti-inflammatory modulators results for the variety Forastero, the content in polyphenols is higher for epicatechin and other procyanidins.
Author Response
Response to reviewer 1
"Please see the attachment"
This paper deals with the evaluation of the antioxidant and immunomodulatory capacities of cocoa Guiana and a comparison with the classic Forastero variety. Moreover, the content of phenolic compounds and methylxanthines were determined by HPLC-DAD. This is the first study on the chemical characterization of this variety of cocoa as well as the bioactivities due to the bioactive compounds. The study was well conducted and may attract the attention of cocoa experts and the scientific community.
Response: Thank you very much for your comments which helped us improve this manuscript. All the document have submit an English revision.
Comments and modifications.
- Abstract: results related to the bioactive compounds determined by HPLC responsible for the bioactivities should be included.
Response: Thanks for your kind suggestion. We included, in the abstract, results related to bioactive compound contents (epicatechin, B2 and C1) before and after fermentation. These bioactive compounds were well-known to be responsible for these biological properties. We add sentences as follows:
“Fermentation altered the cocoa composition by diminishing bioactive compounds which could be responsible for these biological activities. Indeed, after 6 days of fermentation, compounds decreased from 614.1 ± 39.3 to 332.3 ± 29 mg/100 g DM for epicatechin, from 254.1 ± 14.8 to 129.5 ± 20.7 mg/100 g DM for procyanidin B2 and from 178.4 ± 23.5 to 81.7 ± 2.9 mg/100 g DM for procyanidin C1.” (line 25-29)
- Lines 213-215: The sentence should be rewritten because is intricate.
Response: Thanks for your suggestion. Indeed, this sentence is not so easy to understand. We revised it as follows:
“Regarding the extraction with binary solvent (acidified water-acetone, 30/70, v/v), fermentation induced a significant impact. We noticed an increase after 6 days of process from 18.1 ± 0.9% to 20 ± 0.1% for Guiana and from 18.03 ± 0.6% to 20.4 ± 0.5% for Forastero. No varietal difference was noted whatever the process used.” (line 232-235)
- Lines 219-228: The authors should indicate that this data is for unfermented cocoa beans.
Response: Thanks for your kind reminders. Clarification was made for unfermented extracts as follow:
“All results dealing with TPCs and TFCs and antioxidant activities are shown in Table 1.
If no varietal differences were noticed in the TFC assays (308.64 ± 10.1 and 306.5 ± 4.7 µg CE/g DM for unfermented Guiana and Forastero, respectively), a significant varietal difference was noticed in TPCs (41.3 ± 0.8 and 45.9 ± 1.2 mg EAG/g DM for unfermented Guiana and Forastero, respectively).
As in vitro AO activities can’t be determined with a single method, we have selected a few with various AO mechanisms to do so. FRAP induces an electron transfer while DPPH induce a radical electron transfer and ORAC induce a proton-transfer mechanism. Unfermented Guiana presented an equivalent DPPH activity (294.9 ± 17.9 µmol ET/g DM) and FRAP activity (684.6 ± 48.5 µmol Fe2+/g DM) than unfermented Forastero. However, unfermented Guiana presented a 1.3-fold higher activity (1097 ± 111.8 µM ET/g DM) than unfermented Forastero (838.5 ± 67.8 µM ET/g DM) in ORAC assay.
Furthermore, we noticed that fermentation induced a significant decrease in TPCs, TFCs and AO activities for both varieties. Indeed, after 6 days of process, TPCs decreased by 32% in Guiana and 20% in Forastero. Also, Guiana showed a decrease of 25, 31 and 23% for its DPPH, FRAP and ORAC activities, respectively, after 6 days of process. ” (table 1)
- Table 1: I suggest to authors to change the presentation of the results and change the lines by the columns. In this way, the statistical results will be easier to read since will be in the same columns instead of in two blocs of data as presented.
Response: Thank you very much for pointing this out. We have read your suggestion carefully and tried our best to make it more readable. (line 249-250)
- Still for table 1: the authors should be correct the significant numbers of the results. Seams that no criteria have been taken.
Response: Thank you for the nice reminder. We revised the table.
- Table 2: The authors have some explanation for the fact for the lower impact of the fermentation process on the decrease of active compounds? some other components? The chemical or biological reason?
Response: Thanks for your questions. In literature, fermentation is well-known to induce a decrease of methylxanthines (30%) and polyphenols (20%). A part of the explanation can be found in lines 534-549. Indeed, this behavior could be due to various phenomenon as:
- External diffusions through the bean for methylxanthines and polyphenols (Camu et al., 2008)
- Acidic hydrolysis caused by acidic conditions (presence of lactic, citric and acetic acids) (Santander Muñoz et al., 2020)
- Enzymatic degradations (glucosidase and polyphenol oxidase) which impact anthocyanins and flavanol contents by converting them to quinones. In a second time, quinones could complex with other compounds as proteins and other compounds and become insoluble (Santander Muñoz et al., 2020).
When we compare our results with literature, we can note that the impact of fermentation is not lower than other studies. Indeed, fermentation is a complex process which is governed by various parameters as microbial strains or cocoa variety and process variability could also interfere.
- Table 2: Some differences in the content concerning variety can be lost with the statistical analysis performed since the fermentation process has a high impact on the total contents. However, if the authors compare raw beans some differences could be noticed that are related to the variety.
Response: Thanks for your comment. For a better understanding of the data, we used numbers to determine varietal difference and letter to determine the impact of fermentation. Table modified could be found in line 278-279, table 2.
- Al figures: The numbers separator on the graphics should be replaced by a dot (.)
Response: Thank you very much for pointing this out. We went through the entire manuscript and figures to convert the number separator (,) into (.).
- Lines 299-308 and Fig. 4: Since the active compounds are lower? the concentration should be adjusted? Since for Forastero, the differences were not noticed?
The author should include statistical data in these figures.
Response: Thanks for your question. We added statistical data in figures.
- Lines 327-329: If the authors agree, these data could be included as supplementary data.
Response: Thanks for your suggestion, we included them as supplementary data in case they can be helpful for readers.
- Lines 377-387: I suggest to authors eliminate this part of the coffee since data on cocoa beans are available in the literature for discussion, otherwise include other foods also common in the diet.
Response: Thanks for your suggestion. We chose coffee because this matrix seemed relevant to us from the point of view of its relatively close composition to cocoa (high methylxanthine and polyphenol contents). We didn’t include other common foods because it was not reliable for us to compare results that were not operated under our experimental conditions.
- Line 398: "In our case, the TPC of Guiana is different from Forastero." How different? The authors should complete the sentence. In your case is also different, how much x-fold...
Response: Thanks for the comment. We revised the sentence as follow:
“In our case, Guiana TPC was 0.9-fold lower than Forastero (p<0.05) and demonstrated a more efficient ORAC activity (p<0.05) Knowing that TPC assay quantifies a set of polyphenols, this behavior could reflect that Guiana could presented its own quantitative and qualitative distribution in terms of compounds.” (line 490-493)
- Lines 488-494: Theobromine was not considered as a potential bioactive compound, it could be? Some recent studies have demonstrated some anti-inflammatory activity of theobromine. Maybe in the future, the authors should test also authentic standards in order to verify the molecules responsible for the activity.
Response: Thank you. It’s a good question. Indeed, in few studies, theobromine has revealed to be also anti-inflammatory. However, fermentation has affected immunomodulatory activities but not theobromine content. This phenomenon may manifest itself in two hypothesis. Theobromine is not involved in this activity (1) or theobromine participate in an unknown percentage. It is relevant in comparison of other compounds which diminish after fermentation and have a massive impact on this activity?
As the reviewer suggested, using standard will be a good strategy to confirm the responsibility of each compounds in these activities. Our future experiments will be focused on fractions of cocoa extracts to study “bioguiding and synergy effect”. We add in the conclusion our perspective as follow:
“Moreover, study using fractions of compounds could be a good strategy to study bioguiding and synergic effects.” (line 652-653)
- Section 3.4.: Some discussion on anti-inflammatory modulators results for the variety Forastero, the content in polyphenols is higher for epicatechin and other procyanidins.
Response: In our study, we showed that Forastero is as rich in epicatechin and other procyanidins as Guiana. With almost the same TPC, this does not explain a priori the differences in immunomodulatory activities between two varieties. In literature, lack of information concerning variety does not allow results to be found on Forastero type cocoa and make comparison complex. To my knowledge, no studies has presented a comparison of immunomodulatory activities between two cocoa varieties. Does this answer suit to your question?

Reviewer 2 Report
My compliments to the authors because there are just few comments to be considered:
There are just two comments to be considered:
The resolution of all the figures should be improved and statistical analysis should be added.
In the Session Conclusion from line 531 to 541 is not clear why it was necessary to add citations. I suggest to move this part into the discussion session.
Author Response
Response to reviewer 2
Please see the attachment
My compliments to the authors because there are just few comments to be considered.
Response: Thank you very much. We hope our changes have met your expectations.
Comments and modifications.
- The resolution of all the figures should be improved and statistical analysis should be added.
Response: Thank you for your comment. We improved the resolution of all the figures to make them clearer and included statistic data.
- In the Session Conclusion from line 531 to 541 is not clear why it was necessary to add citations. I suggest to move this part into the discussion session.
Response: We transferred and integrated the line 531 to 541 to the discussion session as follow:
“With epicatechin contents of 614 ± 0.39 mg/g DM and procyanidin contents (dimer and strimer) of 512 ± 0.21 mg/g DM, the valorization of Guiana into chocolate could be a good alternative to obtain new sources of antioxidants. For cocoa beans of Guiana, after 6 days of fermentation, we observed a loss of 46% and 49% for epicatechin and procyanidins, respectively. Bordiga et al., 2015 have demonstrated that the transformation of fermented beans to chocolate induced an average loss of epicatechin by 81.8 ± 8.1% for African cocoas [44]. Consequently, if 6 days fermented Guiana was turned into chocolate, we would obtain final epicatechin contents of 1.116 ± 0.071 mg/g DM. Guiana chocolate epicatechin content could be 1.4-fold and 2.4-fold higher than two other dark chocolate contents [44]. The fact that dark chocolates have also demonstrated AO activities in ORAC assay (227 ± 74 µM TE/g DM) [44] suggests that Guiana chocolate could be a good cocoa-derivate with AO capacities.” (line 527-537)
